# Quantifying the Differences between 3D Virtual Planning and Attained Postoperative Reduction on CT for Patients with Tibial Plateau Fractures; a Clinical Feasibility Study

**DOI:** 10.3390/jpm13050788

**Published:** 2023-05-01

**Authors:** Nynke van der Gaast, Jellina M. Huitema, Britt H. J. Edwards, Joy Verbakel, Dagmar S. Alderlieste, Job N. Doornberg, Michael J. R. Edwards, Ruurd L. Jaarsma, Erik Hermans

**Affiliations:** 1Department of Trauma Surgery, Radboud University Medical Center, 6525 GA Nijmegen, The Netherlands; 2Department of Orthopaedic & Trauma Surgery, Flinders Medical Centre, Bedford Park, SA 5042, Australia; 3Department of Orthopaedic Surgery, University Medical Center Groningen, 9713 GZ Groningen, The Netherlands

**Keywords:** tibial plateau fractures, Three-Dimensional Virtual Planning, quantification of measurements

## Abstract

*Background*: Three-Dimensional Virtual Planning (3DVP) has been proven to be effective for limiting intra-articular screw penetration and improving the quality of reduction for numerous fractures. However, the value of 3DVP for patients with tibial plateau fractures has yet to be determined. *Purposes*: The research question of this study is: Can Computed Tomography Micromotion Analysis (CTMA) provide a reliable quantification of the difference between 3DVP and the postoperative reduction on CT for tibial plateau fractures? *Methods*: Nine consecutive adult patients who received surgical treatment for a tibial plateau fracture and received pre- and postoperative CT scans were included from a level I trauma center in the Netherlands. The preoperative CT scans of the patients were uploaded in a 3DVP software. In this software, fracture fragments were reduced and the reduction was saved as a 3D file (STL). The quality of the reduction from the 3DVP software was compared with the postoperative results using CT Micromotion Analysis (CTMA). In this analysis, the translation of the largest intra-articular fragment was calculated by aligning the postoperative CT with the 3DVP. Coordinates and measurement points were defined in the X, Y, and Z axes. The combined values of X and Y were used to define the intra-articular gap. The Z-axis was defined as the line from cranial to caudal and was used to define intra-articular step-off. *Results*: The intra-articular step-off was 2.4 mm (Range 0.5–4.6). Moreover, the mean translation of the X-axis and Y-axis, which was defined as the intra-articular gap, was 4.2 mm (Range 0.6–10.7). *Conclusions*: 3DVP provides excellent insight into the fracture and its fragments. Utilizing the largest intra-articular fragment, it is feasible to quantify the difference between 3DVP and a postoperative CT using CTMA. A prospective study to further analyze the use of 3DVP in terms of intra-articular reduction and surgical and patient-related outcomes has been started by our team.

## 1. Introduction

Thorough understanding of fractures and fracture lines is important to define treatment strategies in orthopaedic trauma. The tibial plateau is one of the key weight-bearing areas of the body; it is exposed to forces five times the body’s weight [1]. Since tibial plateau fractures are always intra-articular fractures and are technically complex fractures to treat, surgeons continue to search for different and innovative ways to address and treat these fractures [2,3,4,5].

Radiographs were primarily used to diagnose tibial plateau fractures and to prepare for surgical treatment. With advancements in Computed Tomography (CT), a more thorough view is provided for the diagnosis and treatment of fractures in general [6]. The indications for surgical treatment of tibial plateau fractures vary between a step-off and/or gap between 3–10 mm [3,7,8,9]. Recently, the study by Vaartjes et al. [7] found that patients who were treated non-surgically with a step-off or gap between 2–4 mm have similar patient-related outcomes compared to patients with a step-off or gap < 2 mm. Their study also concludes that a step-off > 4 mm was associated with more knee symptoms and a lower QOL during follow-up.

The use of 3D-assisted surgery, such as 3D printed models and 3D virtual planning, is a current topic of research in orthopaedic surgery. 3D Virtual Planning (3DVP) software has been developed to provide a multidimensional view of fractures, where the fracture and its fragments can be visualized using advanced bone segmentation. After segmentation, the fragments can be reduced, and implant material can be positioned. The insights 3DVP allows into the fracture and its fragments aim to improve fracture reduction during definitive surgery, shorten operation time and decrease complications such as intra-articular screw penetration [3,5,10,11,12,13]. The systematic review by Assink et al. [12] shows a reduction in operation time, a decrease in the amount of blood loss and also shows a reduction in the number of fluoroscopies required for 3D assisted surgery. Even though the aforementioned studies show promising results, 3D-assisted surgery does take extra time and effort in a surgeons’ work schedule. Therefore, it is vital to assess whether the time and effort required to perform 3D assisted surgery is offset by the quality of the postoperative reduction.

To the best of the author’s knowledge, there is no literature providing quantification for the comparison of 3DVP software to the postoperative CT scan for patients with tibial plateau fractures. Therefore, the research question of this study is: Can CTMA provide a feasible overview of the differences between 3D virtual planning and the postoperative reduction on CT for tibial plateau fractures?

## 2. Materials and Methods

### 2.1. Patient Inclusion

This study was approved by the Medical Ethical Committee (METC) of the Radboud University Medical Center (Ethics approval number: 2021-8231, METC Oost-Nederland). The procedures adhere to the tenets of the Declaration of Helsinki (64 October 2013). Twelve consecutive patients with a tibial plateau fracture who received a preoperative CT scan underwent surgical fixation for <14 days and received a postoperative CT scan according to protocol were included from a level 1 trauma center in the Netherlands (Radboud University Medical Centre, Nijmegen). Patients were excluded when they received previous knee surgery on the fractured knee or had pre- or postoperative CT scans of insufficient quality (Figure 1).

### 2.2. 3D Virtual Planning (3DVP)

The preoperative and postoperative CT scans from all included patients were collected from the hospital’s Electronic Patient Files and were anonymized. The preoperative, axial CT images (Bone setting, slice thickness 0.5 mm) were uploaded on a laptop provided by the 3DVP software developer (Sectra AB © ‘3D trauma for Orthopaedics’, Linköping, Sweden). The software converts axial CT slices into a 3D virtual model that can be freely rotated. The femur, patella and fibula were first marked and removed, creating a model of only the tibia (Figure 2A). All fracture fragments were then segmented, so that each fragment was displayed with a separate color (Figure 2B). The fracture fragments were reduced and after a satisfactory reduction, the implant material was placed (Figure 2C,D). All fractures were first segmented and reduced by four researchers (NG, BE, JV, DA) and the final reduction and placement of implants was analyzed and approved by an experienced trauma surgeon (EH).

### 2.3. Analysis of Postoperative CT vs. 3D Virtual Planning

Computed Tomography Micromotion Analysis (CTMA, version 23.1, Sectra AB ©, Linköping, Sweden) is an interactive software tool developed to detect micromotion between objects such as bones or implants [14,15]. The postoperative CT scan and 3D virtual planning were uploaded into the CTMA software. CTMA requires users to pick a ‘reference body’ and a ‘moving body’. The tibial shaft was used as the reference body and the largest intra-articular fragment of the fracture was used as the ‘moving body’. For bilateral fractures, the two largest intra-articular fragments were analyzed to create an overview of the articular surface on both the lateral and medial side. The ‘reference bodies’ and ‘moving bodies’ were aligned and coordinates and measurement points were defined in the X, Y, and Z-axes. The combined values of X and Y were used to define intra-articular gap. The *Z*-axis was defined as the line from cranial to caudal and was used to define intra-articular step-off. The point of measurement was placed at the center of the joint surface of the largest intra-articular fragments (Figure 3).

Furthermore, a visual comparison of the intra-operative vs. 3DVP was created for all patients. Appendix A present an overview of one patient, where blue represents the virtual reduction and orange represents the reduction of the postoperative CT scan. The measured fragment of this patient was the largest fragment on the medial side.

We reviewed the use of CTMA to analyze the migration of fracture fragments, for this analysis has not been used before analyzing fracture fragments. Measurements were performed twice by the same certified software user to minimize measurement errors. To limit recall bias, the time in between two measurements was two weeks. Furthermore, the intra-observer agreement was calculated for these measurements. In accordance with the divisions of Koo et al. [16], the intra-observer agreement of ≤0.50 was categorized as poor, 0.51–0.75 as moderate, 0.76–0.90 as good and >0.90 as excellent.

## 3. Results

### 3.1. Patient Demographics

Twelve patients received a preoperative CT scan as well as a postoperative CT scan < 14 days after their tibial plateau surgery. For three of the identified patients, the CT images were created by a 3D C-arm in the operating theatres. The quality of this imaging method was insufficient to create a 3DVP and therefore, these patients were excluded. We aimed to include both unicondylar and bicondylar fractures to assess the quality of our quantification for patients who received dual plating. Fortunately, we were able to include six unicondylar fractures and three bicondylar fractures. Patient demographics can be found in Table 1.

### 3.2. 3D Virtual Reduction vs. Surgical Reduction: Analysis of Largest Intra-Articular Fragment

The mean translation on the *Z*-axis, which was defined as the intra-articular step-off, was 2.4 mm (Range 0.5–4.6). Moreover, the mean translation of the X-axes and Y-axes, which was defined as the intra-articular gap, was 4.2 mm (Range 0.6–10.7). The mean translation per patient can be found in Table 2. The differences between the postoperative CT and 3DVP for the patient with the greatest differences and the smallest differences were visualized, where the 3DVP is blue and the post-operative reduction is orange (Appendix A). The patient with the largest differences comparing 3DVP to the postoperative CT was a patient with a large depressed medial fragment, which was classified as a Schatzker 4 (Appendix A). The patient with the smallest differences was a patient with a lateral split fragment that was classified as a Schatzker 2 (Appendix A). The intra-observer agreement for the CTMA measurements was >0.93 and was categorized as excellent.

## 4. Discussion

The aim of this study was to assess the feasibility of CTMA to provide an overview of the differences between 3DVP and the postoperative reduction on CT for patients with tibial plateau fractures. This study shows that a reasonable comparison between 3DVP and a postoperative CT can be created using CTMA for patients with both unicondylar and bicondylar tibial plateau fractures.

### 4.1. Interpretation of Results

In this study, we used CTMA to compare the reduction of the largest intra-articular fragments of 3DVP and the postoperative CT-scan. The mean values show that there were certain differences between 3DVP and the postoperative CT. For example, the intra-articular step-off was less than the intra-articular gap. The effect of intra-articular step-off and gaps on long-term patient related outcomes has been a subject of interest in literature [7,17,18,19]. Since intra-articular step-off and gaps are underestimated on X-rays and even on CT, Assink et. al. created a 3D quantification for measurement of intra-articular step-off and gaps for tibial plateau fractures. Their study showed that patients with a larger intra-articular surface of the tibial plateau fracture on a preoperative CT suffer from poorer long-term patient-related outcomes [20]. However, this quantification was based on the preoperative CT scan. This gap in knowledge has been overcome by a study by Rosteius et al. [21]. Their study showed that a step-off of approximately 2.9 mm and a gap size of 6.6 mm on postoperative CT scans result in acceptable patient-related outcomes. Comparing these results to the differences of respectively 2.4 mm for step-off and 4.2 for gaps, we can conclude that CTMAanalysis can be used to compare 3DVP and the postoperative CT in terms of step-off and gap and can provide a feasible comparison.

Moreover, the intra-observer agreement was calculated for the CTMA measurements to review the use of CTMA for the migration of fracture fragments. We were aware of possible difficulties regarding the measurements of the bilateral fractures in this analysis due to extensive scattering on the postoperative CT scans because of dual plating. The quality of 3D C-arm imaging was not sufficient for this analysis, however, the reductions could be visualized adequately with the use of SEMAR techniques on a postoperative CT scan to remove scattering of the implant materials. With an intra-observer agreement of >0.93 for the measurements, we can conclude that the measurements were reproducible and accurate.

In clinical practice, CTMA analysis could provide surgeons with a quantification of their operative results after tibial plateau surgery. Intra-operative guidance of CTMA is not yet possible in clinical setting. However, integrating CTMA in a mixed-reality platform, could be beneficial for surgeons. Further research on this integration is now pending in our clinic. Moreover, the use of 3DVP in clinical practice to assess outcomes such as differences in operation time, number of fluoroscopies, surgeon’s satisfaction and patient-related outcomes would be a topic of great interest to researchers. To assess the quality of reduction using 3DVP and to assess the additional value using 3DVP for tibial plateau surgery and patient-related outcomes, a prospective study has begun.

### 4.2. Study Limitations

First and foremost, the number of included patients was restricted due to the limited availability of postoperative CT scans for patients with tibial plateau fractures. When analyzing the Schatzker classification of the included patients in this study, 33% were classified as a Schatzker 2, 33% were classified as a Schatzker 4 and 33% were classified as a Schatzker 6. It was noted that Schatzker 4 and Schatzker 6 have a greater representation for our study compared to the incidence reported in literature, where the incidence varies between 12–25% and 8–23% respectively [22,23,24,25]. This is in line with our hypothesis that a surgeon would be more interested in the postoperative reduction for a more complex fracture. For the purposes of our study, this could represent an overestimation of the differences between postoperative CT and 3DVP. Our study shows that regardless of the severity of the tibial plateau fracture, a feasible comparison can be provided using CTMA, despite the relatively small number of analyzed patients. Moreover, we believe that it was relevant to include a representation of both unicondylar and bicondylar fractures, which was achieved with three bicondylar fractures and six unicondylar fractures.

Secondly, when using 3DVP, all fracture fragments are excellently visualized and can be freely moved within the virtual space, without restrictions of soft tissue. However, due to the presence of soft tissue, the smaller fracture fragments are usually left untouched during surgery. Additionally, the 3DVP software allows fragments to be placed on top of each other to create the reduction, without a notification from the 3DVP software, whereas this would not be feasible during surgical fixation.

Thirdly, the comparison between 3DVP and postop CT was calculated based on an axes system with X, Y and Z-values. We defined the Z-value as an accurate representation of the step-off and calculated the squared mean of X and Y as a representation of the gap. Ideally, these measurements would have included the complete surface of the tibial plateau, for example, the quantitative measurements that were proposed by Assink et al. [20]. Our measurements were based on one central measurement point of the largest intra-articular fragments, which we believe was appropriately representative to create the comparison that answered our research question.

## 5. Conclusions

3DVP gives an excellent insight into the fracture and its fragments. Utilizing CTMA, it is possible to quantify the differences between 3DVP and a postoperative CT based on the largest intra-articular fragment. A prospective study to further analyze the use of 3DVP in terms of intra-articular reduction and surgical and patient-related outcomes has begun.

## Figures and Tables

**Figure 1 jpm-13-00788-f001:**
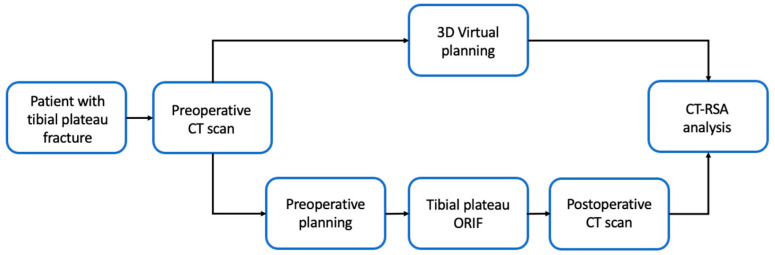
Overview of Study Methods.

**Figure 2 jpm-13-00788-f002:**
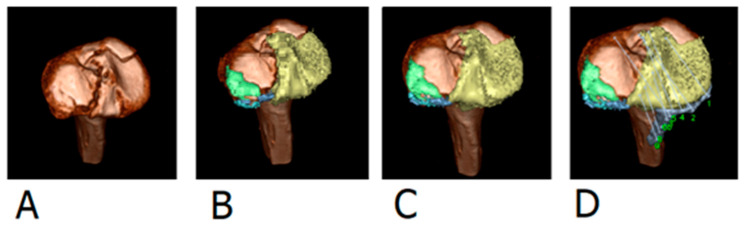
Example of 3D Virtual Reduction: (**A**) Creating a 3D model of the tibia (**B**) Segmentation of each fracture fragment (**C**) Reduction of fracture fragments (**D**) Placement of implant material.

**Figure 3 jpm-13-00788-f003:**
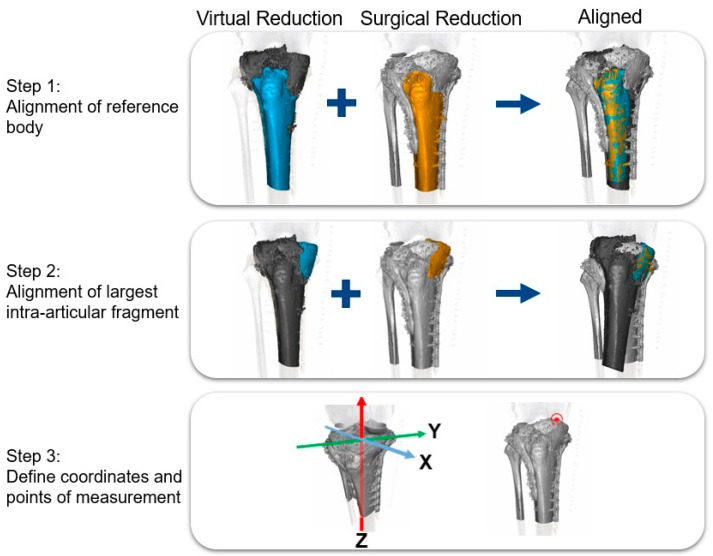
Overview of CT Micromotion Analysis (CTMA).

**Table 1 jpm-13-00788-t001:** Patient Demographics.

Patient Demographics	Total (*n* = 9)
Gender	
Male	4
Female	5
Age (Range)	45.8 (20–69)
Schatzker Classification	
Schatzker I	0
Schatzker II	3
Schatzker III	0
Schatzker IV	3
Schatzker V	0
Schatzker VI	3
Side	
Left	4
Right	5

**Table 2 jpm-13-00788-t002:** Overview of differences between the largest intra-articular fragments on postoperative CT vs. 3DVP.

Patient	Gap (XY)	Step-off (Z)
1	7.0	1.5
2	1.5	0.5
3	10.7	4.6
4	1.4	1.8
5	3.3	1.7
6	3.3	2.0
7	5.7	4.4
8	6.7	3.1
9	0.6	1.5
Mean	4.2	2.3
Range	0.6–10.7	0.5–4.6

## Data Availability

The data used in this will be stored on the department’s server for a period of 15 years. Access to the data is restricted to authorized personnel only. The data will be made available upon request and will be subject to any applicable ethical and legal restrictions.

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
