# Peer review of "Quantifying the Differences between 3D Virtual Planning and Attained Postoperative Reduction on CT for Patients with Tibial Plateau Fractures; a Clinical Feasibility Study"

_jpm, 2023, doi:10.3390/jpm13050788_

Round 1

Reviewer 1 Report

The authors present a feasibility study on the use of 3 D planning in the treatment of tibial plateau fractures. They have listed the small number of cases themselves in the limitations. Postoperative CT control should be standard in the surgical treatment of tibial plateau fractures. The 3 D reconstruction (postoperative) the basis of quality control.

If you would discuss the potential use or the benefit of the comparison of planning and achieved result - this would certainly be beneficial for the manuscript and your work. Is there in principle the possibility of intraoperative support of the surgeon in the sense of a real-time control of the reduction result before osteosynthesis with this method?

Reviewer 2 Report

Preoperative 3D visualization of complex tibial plateau fractures to plan surgical treatment has become standard in traumatology.

In their study, the authors investigate the difference between virtual 3D planning and the actual postoperative CT result using computed tomographic micromotion analysis (CTMA).

It was shown that CTMA can be used to produce a meaningful comparison between 3DVP and postoperative CT for patients with both unicondylar and bicondylar tibial plateau fractures.

Overall, this is an interesting study whose initial question may seem somewhat complicated. 

It would be helpful if the relevance for everyday traumatological practice could be brought to the fore a little more. The technical details, on the other hand, are explained in detail.

Overall, a very interesting study approach with conclusive limitations and the important note that actual results and the associated relevance for the surgeon can only be presented after a prospective study.

I recommend the authors to shorten the technical details somewhat and to emphasize the clinical relevance more.  
